# Multisensor Data Fusion for Localization of Pollution Sources in Wastewater Networks

**DOI:** 10.3390/s22010387

**Published:** 2022-01-05

**Authors:** Krystian Chachuła, Tomasz Michał Słojewski, Robert Nowak

**Affiliations:** Faculty of Electronics and Information Technology, Warsaw University of Technology, 00-665 Warsaw, Poland; tomasz.slojewski.stud@pw.edu.pl (T.M.S.); robert.nowak@pw.edu.pl (R.N.)

**Keywords:** sensors, data fusion, tracking, peak detection, sewage network, IoT

## Abstract

Illegal discharges of pollutants into sewage networks are a growing problem in large European cities. Such events often require restarting wastewater treatment plants, which cost up to a hundred thousand Euros. A system for localization and quantification of pollutants in utility networks could discourage such behavior and indicate a culprit if it happens. We propose an enhanced algorithm for multisensor data fusion for the detection, localization, and quantification of pollutants in wastewater networks. The algorithm processes data from multiple heterogeneous sensors in real-time, producing current estimates of network state and alarms if one or many sensors detect pollutants. Our algorithm models the network as a directed acyclic graph, uses adaptive peak detection, estimates the amount of specific compounds, and tracks the pollutant using a Kalman filter. We performed numerical experiments for several real and artificial sewage networks, and measured the quality of discharge event reconstruction. We report the correctness and performance of our system. We also propose a method to assess the importance of specific sensor locations. The experiments show that the algorithm’s success rate is equal to sensor coverage of the network. Moreover, the median distance between nodes pointed out by the fusion algorithm and nodes where the discharge was introduced equals zero when more than half of the network nodes contain sensors. The system can process around 5000 measurements per second, using 1 MiB of memory per 4600 measurements plus a constant of 97 MiB, and it can process 20 tracks per second, using 1.3 MiB of memory per 100 tracks.

## 1. Introduction

Due to industrialization, society must take care of waste disposal and the efficient operation of sewage treatment plants. Breakdowns in these areas are a growing threat, so it is crucial to continuously monitor utility networks and react as soon as possible. It is achievable to design pollutant tracking systems that use a very simplified model of substance flow. An appropriate detection accuracy is obtained by fusing data from many sensors of different types. Sensor technology constantly improves and, due to developments in the Internet of Things (IoT) space, provided measurements are easier to access for processing than ever.

Researchers’ interest in exploring methods for the detection, localization, and quantification of pollutants in wastewater networks (WWNs) and water distribution systems (WDSs) has grown considerably in recent years, especially regarding sensor technology [1,2,3,4,5,6,7,8,9,10,11,12,13,14,15,16]. Questionable policies regarding waste disposal are prevalent in today’s industry [17,18]. This is why we see more and more incidents connected to WWNs [13,19,20,21].

The characteristics of WWNs and the volatility of pollutants make localization and quantification immensely challenging. The reasons for the difficulty of this task come from the randomness of the injection process involving the type of pollutants, the injection point, the injection amount, time, duration, etc. [22] Moreover, some physical properties of the wastewater are less predictable or more difficult to measure precisely than others.

The need for data fusion in this field is supported by the fact that a simple monitoring system where anomalies are detected independently for each sensor would be lacking at least four desirable features. The first one is marking discharges observed by multiple sensors as more significant than those detected by only a single sensor. The second is the quantification of pollutants by considering the sum of deviations in measurement series connected to a single discharge event. Another one would be the deduplication of alarms by clustering similar observations. The last one is discovering the flow path of the pollutant that leads to easier source localization. Data fusion has also proved to be a viable technique for knowledge discovery in other fields such as bearing fault identification [23].

We propose an enhanced data fusion algorithm for the detection, localization, and quantification of pollutants in WWNs. This article extends the system depicted in Chachuła et al. [24], where we proposed the preliminary and very limited version of the data fusion algorithm. The algorithm presented in Chachuła et al. [24] represents the sewage network as a directed tree. In this structure, there is one and only one path from any point of the sewage network to sink. Our new algorithm represents the sewage network by a directed acyclic graph (DAG), which corresponds to the structure of real WWNs. Therefore, we can now perform fusion in sewage networks presented in the literature.

### Related Work

In the previous algorithm ([24]), pollutant quantification was based on a fixed per-quantity threshold. This assumption is not valid if we observe sensor calibration drift, or the water level in the network changes. Both of these phenomena occur in real sewage networks. In our new algorithm, we use adaptive peak detection.

We also improved our implementation. Even though the current algorithm is more complex than the previous one, we can process 5000 measurements and 20 tracks per second, which is 50 times faster than previously.

Presented improvements broaden possible use cases of the algorithm and bring it closer to being ready for deployment in real scenarios.

Buras and Solano Donado [25] used random forest classifiers to detect and identify pollutants in wastewater and the XGBoost algorithm for predicting the distance from the source to the sensor. However, the proposed models did not quantify the pollutants.

In 2021, Yan et al. [22] proposed an improved genetic algorithm for the pollution source localization in water supply networks. They formulated assumptions about the behavior of pollutants in water, which are also used in this article. Those assumptions were:The pollutant does not react with other substances.The pollutant gradually dilutes with the water flow.The running speed of the pollutant is consistent with the flow velocity of the pipe segment.Pollutants are injected into the network only through the nodes.The probability of injection for all nodes is equal.The sensors can monitor water properties in real-time.

The second assumption does not mean that we estimate the dilution of pollutants. We consider this phenomenon by considering the sum of deviations of samples from the background value in the measurement series. This way, we account for highly diluted targets causing low, long deviations and concentrated ones, high, short deviations.

The assumption about equal discharge probability in every node does not come without drawbacks. There are areas in networks of real cities, such as industrial districts or residential areas known for illegal activities, with higher probabilities of polluting discharges. However, this information may be difficult to access, as it is connected to law enforcement or simply nonexistent, as is the case in artificial networks used in simulations. This article considers a worst-case scenario, where such knowledge is unavailable, and assumes an identical probability of discharge in every node.

In 2013, Sidhu et al. [26] experimented with using microbial and chemical source tracking for the detection of viruses and chemical compounds in urban stormwater runoff. They state the benefits of monitoring the network for human enteric viruses and drugs, which include more reliable public health risk assessments.

In 2016, Yang et al. [27] focused on combating drug abuse by developing a novel sewage sensor for use in wastewater-based epidemiology. They demonstrated that this sensor could be used for the on-site real-time monitoring of wastewater by unskilled personnel.

Varon et al. [28] localized and quantified a large, persistent methane source using coarse and fine satellite instruments. They used a fusion strategy, where the coarse images were used for localization and fine images—for quantification.

Macas and Wu [29] introduced a framework based on an autoencoder for the detection of anomalies in water treatment systems.

In 2020, Montalvo-Cedillo et al. [30] highlighted the significance of combined sewer overflow events in the contamination of receiving bodies such as rivers. They analyzed multiple wastewater parameters to evaluate the behavior of three different sewage discharges into a river.

Jalal and Ezzedine [31] built and compared two models, a support vector machine and a decision tree, to detect anomalies in WDSs. Their evaluation was based on a real dataset retrieved from a water treatment station.

This article’s goal is to describe the improved multisensor data fusion framework, present the results of various evaluation methods, and interpret them to assess the algorithm’s applicability in real-world scenarios.

## 2. Methods

We represent the sewage network by a directed acyclic graph (DAG) G(V,E). Edges e∈E represent wastewater pipes, and nodes v∈V represent buildings and junctions. The direction of edges corresponds with the direction of the water flow. In some nodes, there are sensors measuring different properties of wastewater. The sensors could be the same type (measure the same property) or different types. Measured properties of wastewater are referred to as entities. Introducing some amount of a pollutant (target) to a node is called a discharge event. Such an event could cause peaks in measurement series provided by sensors along the flow paths if the event refers to the compound that changes a property that is being measured by the sensor.

Edges have two variable attributes: $o_{e}$, the time needed for the water to flow through the whole pipe, and $g_{e}$, which is the dispersion factor. The dispersion factor is calculated by dividing the expected peak height at the end of the pipe by the expected peak height at the start of the pipe.

The data fusion algorithm is a loop of six steps: resampling, peak detection, pollution quantification, downstream propagation, tracking, and event generation. See Figure 1. It processes measurements in real-time, and the steps presented in Figure 1 are repeated for successive, equally spaced points in time as dictated by the sampling period.

Our algorithm supports sensors located in different places and sampling at different rates. The algorithm’s first step (resampling) converts sensor output (measurements) into a unified discrete-time domain. This conversion is achieved either by mean aggregation (if there are several sensor samples in the single algorithm period) or linear interpolation (if a sensor samples slower than the algorithm period). This step’s details and mathematical formulas were presented in Chachuła et al. [24].

The peak detection algorithm classifies measurements into either peaks or background. This step filters sensor output (measurements), to reduce the amount of needed computation. We present the details of this step in Section 2.1.

In the pollution quantification step, measurements marked by peak detection as significant are analyzed further to approximate amounts of pollutants. This step transforms observations in the measurement domain (sensor, entity, time, value, uncertainty) into observations in the target domain (target, node, time, amount, confidence) called detections. Our method, that extends the approach presented in Chachuła et al. [24], is shown in detail in Section 2.3.

Downstream propagation approximates the amount of substance (pollutant) in every node, even if the node has no sensor. This step also smooths out the tracks, which are our estimation of the state of a pollutant flowing through the network. The tracking algorithm, uses the Kalman filter, which finds tracks’ state by clustering detections. Downstream propagation was explained in Chachuła et al. [24], and tracking is extended as presented in Section 2.2.

Event generation creates potential events based on tracks. It is the final step of the algorithm, and its primary purpose is to report only tracks with a large number of supporting (associated) detections. For such tracks, an event represents the discharge of a compound into a graph node. This step focuses the user on essential changes in the sewage network. Event generation is depicted in detail in Chachuła et al. [24].

### 2.1. Adaptive Peak Detection

The peak detection module classifies measurements into three classes: background, up peaks, down peaks. The background class contains measurements of value similar to other measurements from the same series. The up peak class is a set of consecutive measurements of value that stands out from the background in the positive direction and down peaks—in the negative direction.

The module’s task is to detect measurements that belong to peaks. Detected peaks are used to determine the time of discharge events, and the amount of substance discharged is calculated from the peak area. The implemented algorithm is a modification of an algorithm created by Palshikar [32]. We modified it to work with streams of measurements and to detect both up peaks and down peaks.

In our previous solution [24], we did not use adaptive peak detection. The pollutants were detected by comparing measured values to a fixed background level. If the deviation from the background was greater than a fixed threshold, target presence observations (detections) were created based on this measurement. Such an approach works only if the background level is constant and signal to noise ratio is very high. However, in real conditions, the background level varies with time, and currently available sensor technology produces measurements with high noise levels [1]. Using such fixed thresholds in real scenarios was difficult, as the characteristics of real measurements are complex. We created an adaptive peak detection module to enable the data fusion algorithm to produce correct results in real cases.

We denote the set of measurements belonging to a single series as $M=\{x_{1},x_{2},\ldots ,x_{N}\}$, where $x_{i}$ is a value measured at the discrete time *i*. The module works with streams of measurements. This means that, at the moment of analyzing a measurement from time *j*, we only know the values of the measurements $M_{j}=\left\{x_{1},x_{2},\ldots ,x_{j}\right\}$. Palshikar ’s algorithm is based on assigning a peak score to each of the measurements. This score is the value of function $S(M_{j})$. In this algorithm, it is proposed that each point for which $S(M_{j})⩾\theta$ is considered a peak, where θ is some predetermined or calculated threshold.

However, we are looking for both positive and negative peaks and only store a limited number of measurements from the past. The assumption that for $S(M_{j})⩽\theta$ the point is part of up peak and for $S(M_{j})⩾-\theta$ it is part of a down peak, could in many cases cause detection of a false up peak after the occurrence of a true down peak or detection of a false down peak after the occurrence of true up peak. For this reason, we calculate two functions $S_{+}\left(M_{j}\right)$ and $S_{-}\left(M_{j}\right)$. The first one is the score of an up peak, and the second one is the score of a down peak. It may happen that at the same time, both $S_{+}\left(M_{j}\right)⩾\theta_{+}$ and $S_{-}\left(M_{j}\right)⩽\theta_{-}$. In such a case, such a point could be classified as a component of both an up peak and a down peak.

To solve this problem, it is not enough to take the greater of $|S_{+}\left(M_{j}\right)|$ and $|S_{-}\left(M_{j}\right)|$. If such a solution was used, a similar problem would be as in the case of using one $S(M_{j})$ function. Therefore, each of the measurements is given the attribute $p\in \left\{-1,0,1\right\}$ (Algorithm 1 lines 5–11).
(1)$$p\left(x_{j}\right)=\left\{\begin{array}{cc} -1, & \mathrm{if}\;S_{-}\left(M_{j}\right)⩾\theta_{-}\;\mathrm{and}\;S_{+}\left(M_{j}\right)<S_{-}\left(M_{j}\right)\\ 1, & \mathrm{if}\;S_{+}\left(M_{j}\right)⩾\theta_{+}\;\mathrm{and}\;S_{+}\left(M_{j}\right)⩾S_{-}\left(M_{j}\right)\\ 0, & \mathrm{otherwise} \end{array}\right.$$

If we divide the set of all measurements into subsets $T_{l}=\left\{x_{i},x_{i+1},\ldots ,x_{i+m}\right\}$ such that $p\left(x_{i}\right)=p\left(x_{i+1}\right)=\ldots =p\left(x_{i+m}\right)$ and at the same time $p\left(x_{i-1}\right)\neq p\left(x_{i}\right),p\left(x_{i+m+1}\right)\neq p\left(x_{i}\right)$. As the peak consists of two slopes, we merge two consecutive sets of points $T_{l}$, $T_{l+1}$ such that $t\left(T_{l}\right)=\left\{-1,1\right\}$ and $t\left(T_{l+1}\right)=\left\{-1,1\right\}$ in such a way that each of the sets $T_{l}$ is linked to at most one other set.

So if the currently analyzed measurement $x_{j}$ belongs to $T_{l}$ and $T_{l-1}$ has not been combined with $T_{l-2}$, then if $r(T_{l})=1$, and $r(T_{l-1})=-1$, then the set $T_{l-1}\cup T_{l}$ is classified a down peak, because there is a descending slope first and then a rising slope, and if $r(T_{l})=-1$, $r(T_{l-1})=1$, then the set $T_{l-1}\cup T_{l}$ is classified as an up peak because it contains the rising slope first and then the descending slope. Function $r(T_{l})$ is value of function $p(x_{i})$ for each $x_{i}\in T_{l}$.
**Algorithm 1:** Peak detection.
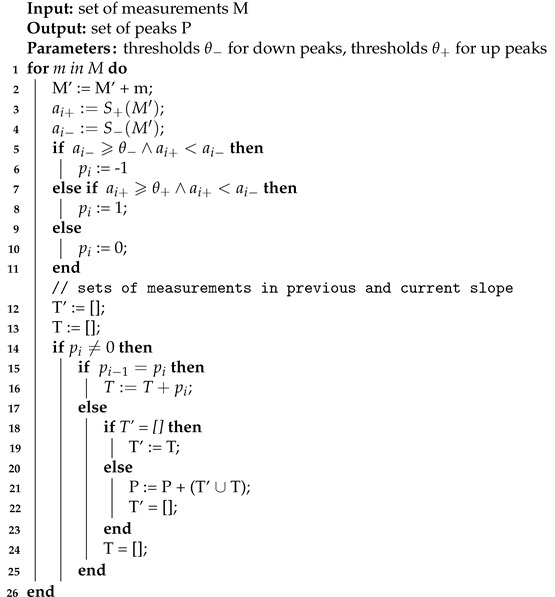


The value of the thresholds $\theta_{+}$ and $\theta_{-}$ could be constant. However, we applied a solution in which they adapt to measurement values. Their values depend on the standard deviation of the function values $S_{+}\left(M_{j}\right)$ and $S_{-}\left(M_{j}\right)$. We used thresholds [32] adapted to the problem of detecting both up peaks and down peaks.
(2)$$\begin{array}{cc} \theta_{+} & =\left(h\cdot s_{+}^{\prime }\right)+m_{+}^{\prime } \end{array}$$
(3)$$\begin{array}{cc} \theta_{-} & =\left(h\cdot s_{-}^{\prime }\right)+m_{-}^{\prime } \end{array}$$

$s_{+}^{\prime }$ is the standard deviation of function $S_{+}\left(M_{j}\right)$, and $m_{-}^{\prime }$ is the mean of this function. Similarly, $s_{-}^{\prime }$ and $m_{-}^{\prime }$ are the standard deviation and the mean of function $S_{-}\left(M_{j}\right)$. *h* is a user-specified positive constant that determines the sensitivity of detection. The smaller *h* is, the more sensitive the algorithm is.

Our solution works with stream measurements. We calculate the standard deviation and mean using only a limited number of past measurements. To calculate the standard deviation, we used the Welford algorithm [33], designed to calculate the approximate value of the standard deviation for data streams.
(4)$$s_{n}=s_{n-1}+\left(\frac{n-1}{n}\right){\left(x_{n}-{\bar{x}}_{n-1}\right)}^{2}$$

An important element of the entire algorithm is the peak score function. The function we use is the maximum difference between the value of the analyzed measurement [32] and the values of the previous *k* measurements, where *k* is a user-specified parameter of the detection algorithm.
(5)$$\begin{array}{cc} S_{+}\left(M_{j}\right) & =max\left(x_{j}-x_{j-1},x_{j}-x_{j-2},\ldots ,x_{j}-x_{j-k}\right) \end{array}$$
(6)$$\begin{array}{cc} S_{-}\left(M_{j}\right) & =max\left(x_{j-1}-x_{j},x_{j-2}-x_{j},\ldots ,x_{j-k}-x_{j}\right) \end{array}$$

A low value of the parameter *k* favors the detection of peaks of short duration.

### 2.2. Tracking in DAGs

The aim of tracking is to follow the state of pollutants. When a pollutant is discharged into the network, it starts flowing in the direction of edges starting in the discharge node. During tracking, detections along the flow paths are clustered so that these caused by a single discharge event are part of a single track cluster.

We use the Kalman filter in the tracking algorithm. The Kalman filter is typically used in metric space, such as $R^{3}$. Our algorithm does not use geographical positions of sensors and pollution because we represent sewage networks by DAGs. The average position of pollution used in tracking is a real number (one-dimensional position), defined as the distance from the start point of the track. We cannot use the distance to a sink node because there is no guarantee that a single sink exists in a DAG. Therefore we represent the target position as a pair of two values: the starting node and time offset. The starting node is the node where the track was created. The time offset is the sum of time offsets on the path between the starting node and the node where the target was last observed.

In our previous algorithm [24], when we only considered networks that are directed trees, we represented positions as a single number, the distance from the sink. The current solution allows representing all possible flow paths from the starting node. This is necessary in DAGs that can contain nodes with more than one outgoing edge. The Kalman filter state contains position and the first derivative of the position. The filter is two-dimensional and has improved the precision of track modeling.

First, let us focus on modeling a track as a linear dynamic system. We assume velocity $\dot{d}$ of the track is constant with the possibility of correction via measurements. The decision to model this velocity as a constant is supported by the fact that pollutants are observed only in network nodes. This results in short bursts of observations separated by long periods with no observations. The constant velocity with high observation noise covariance results in linear interpolation of the track’s position between nodes.

The position *d* changes according to the wastewater velocity. This relationship is presented in Equation (Equation 7).
(7)$$\begin{array}{cc} {\dot{d}}_{k+1} & ={\dot{d}}_{k} \end{array}$$
(8)$$\begin{array}{cc} d_{k+1} & =d_{k}+{\dot{d}}_{k} \end{array}$$

A general equation representing the state of a linear dynamic object without input or offsets but with transition and observation noise is presented in Equation (Equation 9). $z_{k}$ is the object’s output, or in this case, the position *d* of the track as mentioned above. *Q* and *R* are transition covariance and observation noise covariance, respectively.
(9)$$\begin{array}{cc} x_{k+1} & =Ax_{k}+\epsilon_{k+1}^{1} \end{array}$$
(10)$$\begin{array}{cc} z_{k} & =Cx_{k}+\epsilon_{k}^{2} \end{array}$$
(11)$$\begin{array}{cc} \epsilon_{k+1}^{1} & ∼N(0,\;Q) \end{array}$$
(12)$$\begin{array}{cc} \epsilon_{k}^{2} & ∼N(0,\;R) \end{array}$$

Let $x_{k}$ be the state of a track.
(13)$$x_{k}=\left[\begin{array}{c} d\\ \dot{d} \end{array}\right]$$

Then Equation (Equation 7) can be rewritten as Equation (Equation 9) with the following transition matrix *A* (Equation (Equation 14)) and observation matrix *C* (Equation (Equation 15)). The observation matrix indicates that only *d* can be directly observed.
(14)$$A=\left[\begin{array}{cc} 1 & 1\\ 0 & 1 \end{array}\right]$$
(15)$$C=\left[\begin{array}{cc} 1 & 0 \end{array}\right]$$

We propose that the state vector is extended by adding first-order derivatives to this vector (Equation (Equation 13)). The track’s state is not constant. Its position changes at a constant rate equal to the velocity of wastewater. Such a modification causes the filter to represent the state of tracks more accurately.

### 2.3. Better Amount Approximation with Flow Rate

The flow rate of wastewater has an impact on detected amounts of pollutants. Due to dispersion [25], in high flow rate conditions, the peaks are smaller and wider. This means that a larger part of the peaks is indistinguishable from noise, causing quantification error to increase. We wanted to take this effect into account when approximating amounts of detected targets. We introduced an additional dynamic property of the network’s nodes—attenuation. Attenuation *c* is a real number that indicates how much smaller detected amounts $\hat{a}$ in this node are in comparison to the real expected amounts *a*. This relationship is presented in Equation (Equation 16).
(16)$$\hat{a}=c\cdot a;\;c\in \left(0,1\right]$$

The equation can be rearranged to calculate actual amounts based on detected amounts like in Equation (Equation 17) This is used in the quantification step after applying the amount function [24].
(17)$$a=\frac{\hat{a}}{c}$$

## 3. Results

In order to evaluate our improved algorithm, we conducted a set of experiments. Those experiments were based on three different sewage networks. First one we called the *diamond* network (Figure 2a). It is a simple artificial DAG that we included to show scenarios occurring only in the DAG and not in the graphs, which are trees. In such networks, the single discharge event creates more than one pollutant (of the same type) that follows different paths. The second network is the *city* network (Figure 2b). It is a sewer network of an undisclosed sub-catchment area of a European city. The same network was used by Buras and Solano Donado in [25]. We included this network to compare our methods to those presented in their work and to provide a realistic simulation scenario by using a real network. The third network is the example network number 2 included with the EPANET software (Figure 2c). Please note we used a modified EPANET network in our previous work [24]. This network had to be simplified to be a tree by removing a small number of edges. Now, we use this network in its original form. See Appendix A for detailed renders of these networks.

### 3.1. Exponential Signal Generator

In numerical experiments based on simulated data, we use a collection of signal generators. A signal generator generates a single measurement series. A measurement series is a stream of measurements of a given entity in a given node. The signal generator stores information such as the peak value, area, delay, and internal state.

After analyzing measurement series from a real sewage network, we concluded that an exponential function could be a good approximation of a real signal. Similar approach was used in Yan et al. [22]. We provided an exponential signal generator that use function f(k) defined in Equation (Equation 18). The exponential peak is a result of a single discharge event.
(18)$$f\left(k\right)=a\times b^{k};\;k=0,1,2,\cdots \wedge b\in \left(0,1\right)$$

Based on the method presented in [24], the sum of deviations from the background value represents the amount of compound discharged. Considering the whole peak (as k→∞), we can calculate this sum as presented in Equation (Equation 19).
(19)$$\sum_{k=0}^{\infty }\left|f\left(k\right)\right|=\frac{\left|a\right|}{1-b}$$

The $b_{min}$ (Equation (Equation 20)) is a value of *b* that guarantees the sum of deviations is at least *A*, assuming a fixed *a*. This means that $b\in [b_{min},1)$ can be chosen arbitrarily to achieve the desired curvature.
(20)$$b_{min}=1-\frac{\left|a\right|}{A}$$

After choosing *b*, the generator needs a number of samples *N*, after which the desired sum *A* is achieved. Using Equation (Equation 21), we can calculate *N* as seen in Equation (Equation 22).
(21)$$A=\sum_{k=0}^{N}\left|f\left(k\right)\right|=\frac{\left|a\right|\left(b^{N+1}-1\right)}{b-1}$$
(22)$$N=\log_{b}\left[\frac{A}{a}\left(b-1\right)+1\right]-1$$

### 3.2. Success Rate

We measured success rate by running multiple random simulations with a variable number of sensors on every network and counting those, where the result matched the discharge node and compound. We performed 20,000 simulations using all three networks depicted in Figure 2a–c. The results are presented in Figure 3.

The success rate depends tightly on the number of sensors present in the network. We observed a one-to-one relationship between the network coverage and the success rate. This result corresponds with previous findings in Chachuła et al. [24], but this time we performed a lot more simulations to get more precise results.

### 3.3. Distance Error

To assess the accuracy of event localization, we measured the distance (in meters) between nodes pointed out by the fusion algorithm and nodes where the simulated discharge was introduced. The results for each network are presented in Figure 4.

Localization error is highly sensitive to network coverage. An important observation is that, regardless of the network, we achieved the median of zero error when sensors were present in more than half of the nodes. This is a significant improvement from the results presented in Buras and Solano Donado [25], where the median distance error in the city network ranged from 4 to 11 m and sensors were present in every node of the network.

### 3.4. Sensor Success Rate

We introduced a metric of the importance of specific sensor locations. The following plots were created by running multiple simulations with random sensor placement and random discharge nodes (we used uniform distribution here). Every node was assigned a number, which is a fraction of all simulations with a sensor in a given node, where event reconstruction was successful. The results for each network are presented in Figure 5.

For the *diamond* network (Figure 5a) 100% success rate is achieved when a sensor is present in node n4 (node labels are presented in Figure 2a) because all of the wastewater flows through this node. Sensor readings in nodes n2 and n3 are not influenced by discharges into node n4, as they are upstream. The consequence of this is that sensors in nodes n2 and n3 have around 90% success rate. Sensors in node n1 have lowest success rate equal to 84%. The reason for this is that sensors in this location are able to detect discharges only in node n1.

In other networks we can see similar trends. The final sink–nodes are able to detect all of the discharges. The further upstream, the lower the success rate, because of the fact that sensors can only observe discharges that happen upstream from them. Looking at Figure 5a,b we can see irregularities, which are caused by the greater number of nodes requiring more random simulations to achieve smooth and clear results. An important remark is that in these two networks, sensor importance never dropped below 88% and its dispersion is low. This result means that in real networks, decisions about sensor placement should rely on additional knowledge such as the probability of discharge in different nodes.

### 3.5. Sensor Distance Error

An additional sensor importance metric is the sensors’ share in the overall distance error. This was measured by computing the average non-zero distance between the actual discharge node and the reconstructed discharge node. The average was calculated for every sensor that was present in a given simulation. The results for each network are presented in Figure 6.

The maps of sensor importance and distance error highlight the preferred location of sensors. In reality, it is practical to deploy only a few sensors in the network. Extensive simulations can hint at important locations to aid in optimal sensor placement. Our simulations indicate that sensor placement is always a trade-off between success rate and distance error. Sensors placed far downstream have a large chance of detecting discharges, as they are more often on the flow path of the pollutants. Sensors placed further upstream minimize distance error with a cost of success rate.

### 3.6. Peak Detection

We ran the peak detection on a variety of measurement series. We tested the module’s operation on both real data (Figure 7) and simulated data (Figure 8, Figure 9 and Figure 10). Real measurements were taken by Micromole sensors [1], and simulated measurements were generated by the aforementioned exponential signal generator.

In Figure 8, we can see that the algorithm correctly classifies imperfect peaks with emerging outliers. Moreover, the module can cope with different peak heights (Figure 10a) and when both types of peaks are present in a single series of measurements, as is presented in Figure 10b.

The peak detection module adapts to the characteristics of the measurements. The sensitivity of the peak detection is lower when the variance of the measurements is greater. Moreover, we observed that the peak detector could not cope with outliers when they are in the initial phase of a peak. In this case, a single peak is treated as two separate peaks.

We developed the presented algorithm to work online and analyze current data from sensors. The algorithm represents the state of the sewage network for a given time point and produces the most probable events. Such assumption eliminates the possibility of analyzing the time series from sensors offline (using historical data) and then finding the events using optimization algorithms, e.g., by simulating many times the network states and comparing the simulation results with measured values ([22]).

### 3.7. Attenuation in Quantification

To assess how errors in attenuation data influence the pollutant amount error, we ran multiple random simulations with Gaussian disturbance to the attenuation attribute. Figure 11 depicts this relationship. We can observe that the greater the disturbance, the greater the error, which is expected. Looking at the scale of the error, we concluded that the accuracy of target amounts is very sensitive to errors in attenuation. However, the approximation of detected amounts is rough in the first place, as the behavior of compounds in wastewater is hard to model. For the purposes of this study, achieving quantification error within an order of magnitude is considered a success.

### 3.8. Multipath Tracking

The characteristics of directed acyclic graphs can cause an apparent doubling of events in certain cases. If the *diamond* network (Figure 2a) is considered, there are two paths of different lengths between nodes *n1* and *n4*. The first path: *n1* → *n2* → *n4*, has length equal to 120 and the second path: *n1* → *n3* → *n4* has length equal to 157. This means that if a discharge event happens in *n1* at time *k*, then five peaks will be observed in the network: in *n1* at *t*, in *n2* at t+60, in *n3* at t+90, in *n4* at t+120 and in *n4* at t+157. An example of this behavior can be seen in Figure 12.

This event doubling can make tracking difficult if two events happen in succession. In such cases, choosing a correct clustering threshold is crucial. See Figure 13, where a threshold of 20 s was used.

In our opinion, working online and generating the most probable set of events using available information main advantage is allowing to react and alarm as soon as possible. However, there is a risk of false alarms. Moreover, the set of events can change when new information is available.

In the literature, the flow in sewage and water network is modeled by Poisson distribution [22]. Such distribution is achieved when we assume the single household uses the network as a rectangular pulse process with random time and intensity. In the real data available for as this is true, we also observe Poisson distribution in our systems; however, in our algorithm, we decide not to model the flow in the network. We just use measured data from sensors and we assume that the flow is provided by the sensors on an ongoing basis and is constant within the calculation cycle. This assumption simplifies the model. We consider this assumption as valid because the measuring cycle is short.

### 3.9. Resource Usage

The performance and resource usage were evaluated by running a simulation with a variable number of significant measurements and a constant number of tracks. These experiments show how quickly and efficiently the algorithm processes measurements. Tests were run on a fourth-generation Intel Core processor. In Figure 14a, we can see that, on average, the system can process around 5000 significant measurements per second with a constant component of 1 s. We also measured the maximum resident memory of the process (Figure 14b), which stabilizes at around 1 MiB per 4600 significant measurements plus a constant of 97 MiB.

We also repeated these experiments with a constant number of significant measurements and a variable number of tracks. In Figure 15, we presented the results. The relationship between the number of tracks and CPU time is linear. In this scenario, the CPU time required for processing additional 20 tracks is approximately 5 s. Memory requirements for additional tracks are approximately 1.3 MiB per 100 tracks.

Resource usage was significantly improved compared to the previous iteration of the algorithm in Chachuła et al. [24]. Adjusting implementation details caused the system to be able to process measurements 50 times faster. The memory requirements were previously not measured directly, but we consider the result of 1 MiB per 4600 significant measurements a success for a proof-of-concept system. The constant 97 MiB overhead is notable but practical compared to the memory available in modern systems. We calculated and then observed that the time and memory complexity of the proposed data fusion algorithm is linear with respect to the number of detections and tracks. However, memory complexity with respect to the number of tracks is problematic to observe, as each track introduces a very small memory overhead. At this scale, processes such as garbage collection play a significant role.

## 4. Discussion

Resampling introduces sampling error if measurements are not periodic or the period is different from the sampling period. We developed our algorithm to detect pollution in the sewage network, and according to domain experts, we should process the changes of measured values every minute. Our system calculates the sewage network state every second. Therefore, the resampling error is negligible. We assume that the period will be small enough to introduce a minimal error in real scenarios. Moreover, we used simulated data, and we assumed that the generated measurements are periodic, and the period is equal to the sampling period.

All data included in the measurement domain are provided by sensors. We do not use uncertainty in the current version of the algorithm. However, such information can be used in future work to calculate the detection confidence coefficient that we already use in the event generation step of the algorithm.

In our approach, we do not model pollutant spreading. We require a short list of all possible pollutants. In the quantification process, each pollutant is considered for every measurement. Finally, the best pollutant is reported. In recent years, especially for air pollutants, hierarchical methods that model spatio-temporal processes and measurement noise were popular [34,35]. We plan to apply such models in the future to make our results more accurate.

The success rate and distance error of specific sensors indicate that there is a trade-off between those two metrics when considering sensor placement. However, the correctness of the results still strongly depends on how many sensors are present in the network and their placement. Another limitation of the proposed solution is the reliance on high-quality data. Simulated measurements are the best-case scenario. Data from real sensors are of varying quality because of the sensor technology and the characteristics of the wastewater networks.

Further research should focus on evaluating the data fusion algorithm in real-world scenarios, which means running fusion using data from real sensors in live wastewater networks. This will require defining a procedure for choosing the best values of the parameters of the algorithm. Parallel processing techniques could be applied to the proposed solution in case current performance is not enough in large networks with many sensors. Investigation of the influence of sensor placement and discharge amount on the uncertainty of reconstructed events could improve the calculation of event confidence coefficients. Similarly, the uncertainty of measurements, as reported by sensors, could be considered when calculating these coefficients.

## 5. Conclusions

The proposed improvements to the data fusion algorithm significantly broadened its applicability. The new, adaptive peak detection algorithm can adapt to changing background values as presented in Section 3.6. It is a milestone for testing this system in real-world networks. Implementing tracking in DAGs poses a significant improvement from previous achievements as real-world networks no longer have to be modified to perform fusion in as shown in Section 3.8. Results in Section 3.7 suggest that accounting for flow rate by introducing the attenuation coefficient helps when detected pollutant amounts are consistently low because of a high flow rate. Unlike the previous iteration, these improvements make the proposed multisensor data fusion algorithm applicable in real-world scenarios.

Moreover, the development mentioned above did not negatively influence the correctness of the results. This was verified by measuring success rate the same way as in our previous study, using the same network, this time unmodified, in Section 3.2. We performed more simulations to clearly show the linear relationship between the number of sensors and the success rate. In order to evaluate the algorithm in more depth, additional metrics were calculated.

The applicability was not the only aspect of the algorithm that was improved. Simplifying the implementation caused it to be able to process measurements 50 times faster than the previous one as presented in Section 3.9. The measured processing speed and memory requirements show that such a system could run on a regular computer and monitor the network in real-time.

## Figures and Tables

**Figure 1 sensors-22-00387-f001:**
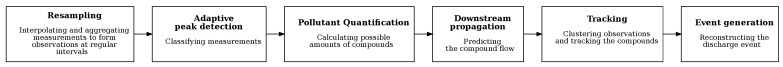
The data fusion algorithm.

**Figure 2 sensors-22-00387-f002:**
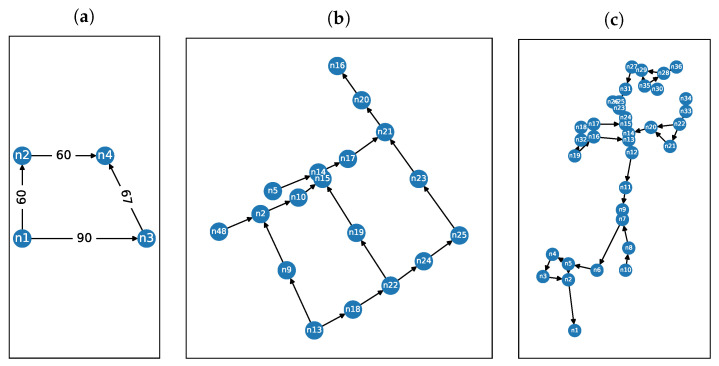
Networks used in experiments: (**a**) the *diamond* network, (**b**) the *city* network, (**c**) the *EPANET* network.

**Figure 3 sensors-22-00387-f003:**
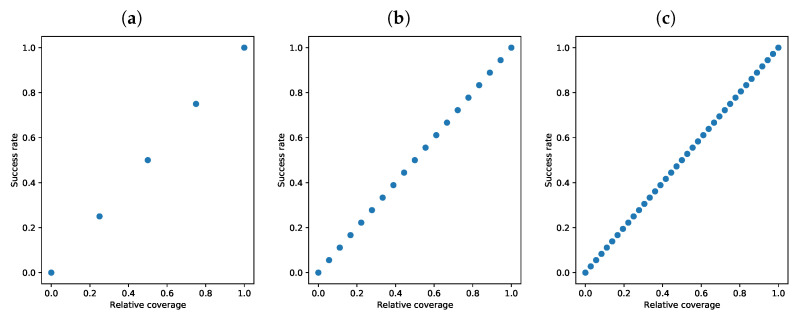
The success rate depends on the network coverage. (**a**) *diamond* network; (**b**) *city* network; (**c**) *EPANET* network.

**Figure 4 sensors-22-00387-f004:**
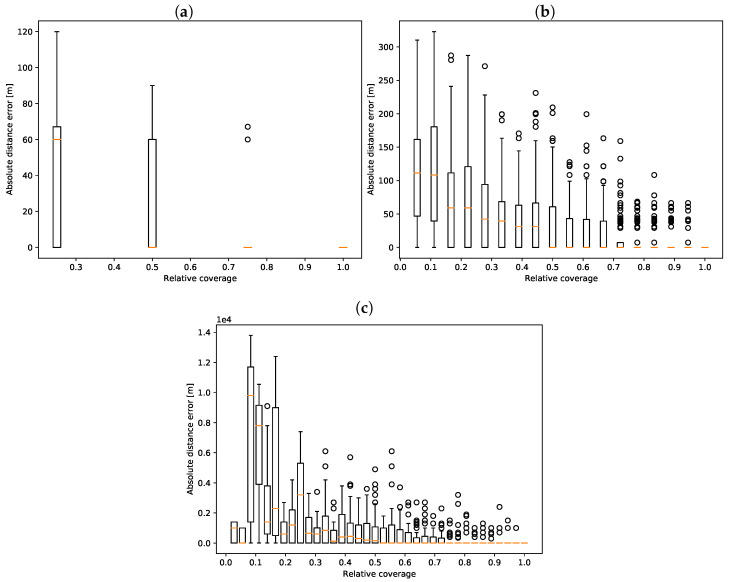
Distance error depends on the network coverage. (**a**) *diamond* network; (**b**) *city* network; (**c**) *EPANET* network.

**Figure 5 sensors-22-00387-f005:**
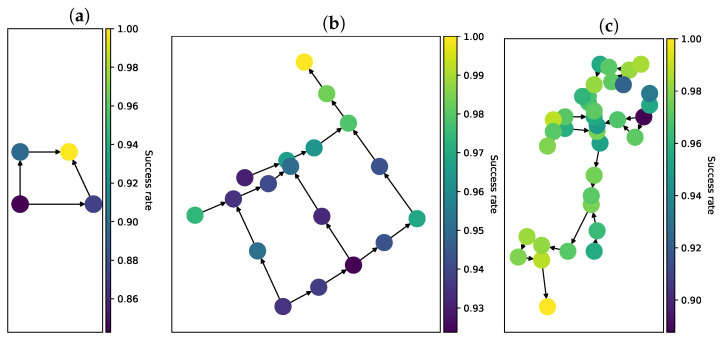
The success rate of sensors. (**a**) *diamond* network; (**b**) *city* network; (**c**) *EPANET* network.

**Figure 6 sensors-22-00387-f006:**
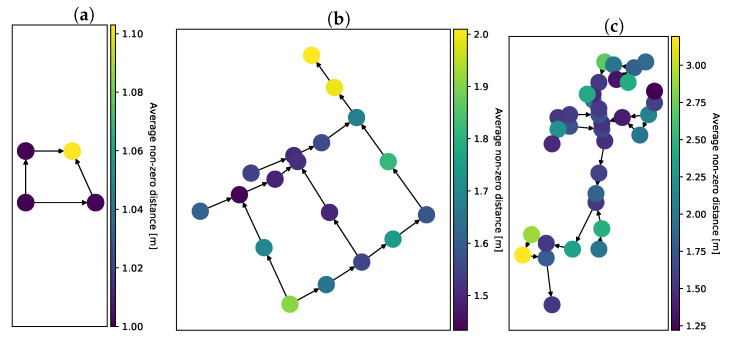
Sensors’ share in localization error. (**a**) *diamond* network; (**b**) *city* network; (**c**) *EPANET* network.

**Figure 7 sensors-22-00387-f007:**
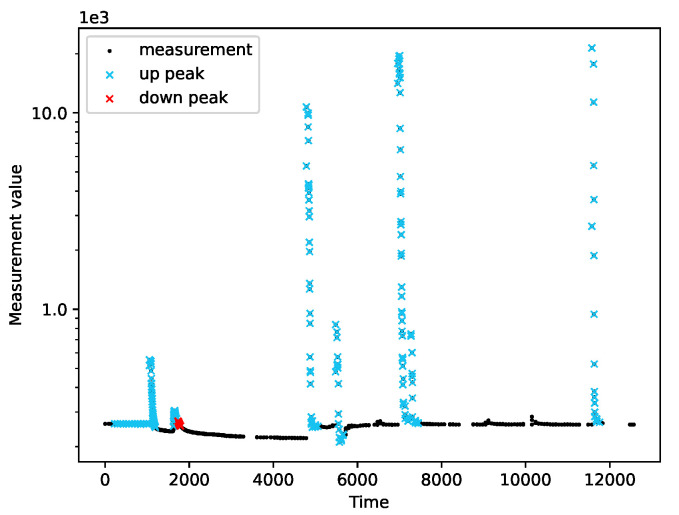
The output of the peak detection algorithm for real measurements taken by Micromole sensors.

**Figure 8 sensors-22-00387-f008:**
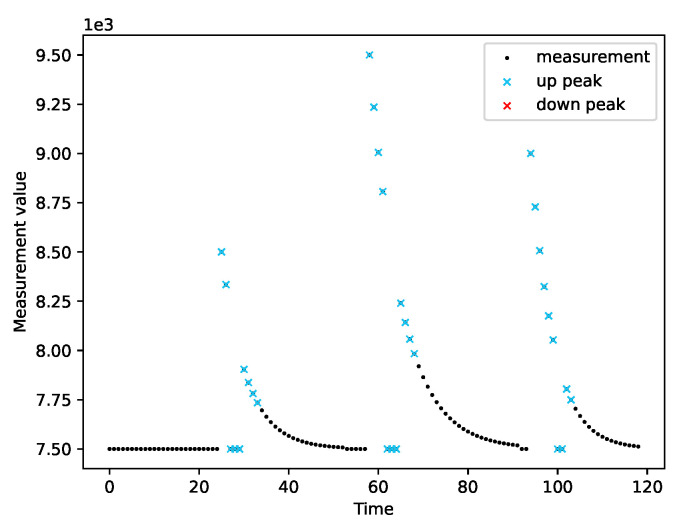
The resistance of the peak detector to the occurrence of outliers.

**Figure 9 sensors-22-00387-f009:**
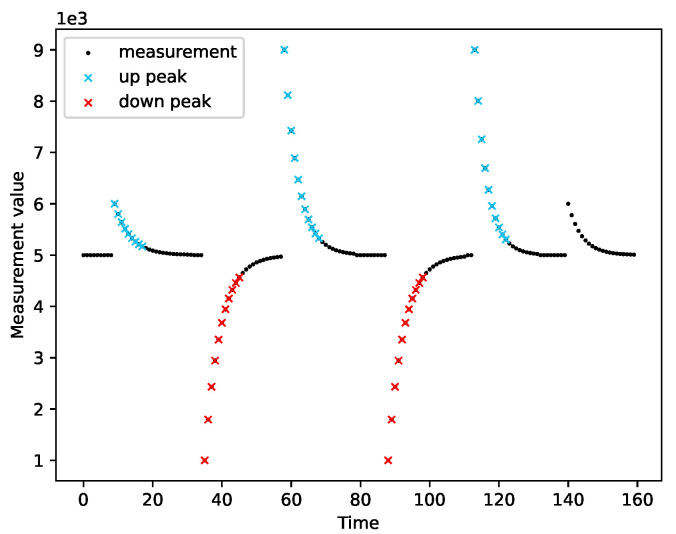
Correctness of peak detection for a measurement series with large variance.

**Figure 10 sensors-22-00387-f010:**
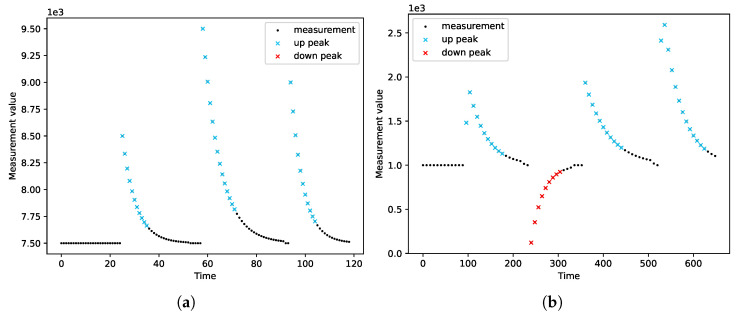
Detection of different types of peaks. (**a**) Different heights of up peaks; (**b**) Both up peaks and down peaks.

**Figure 11 sensors-22-00387-f011:**
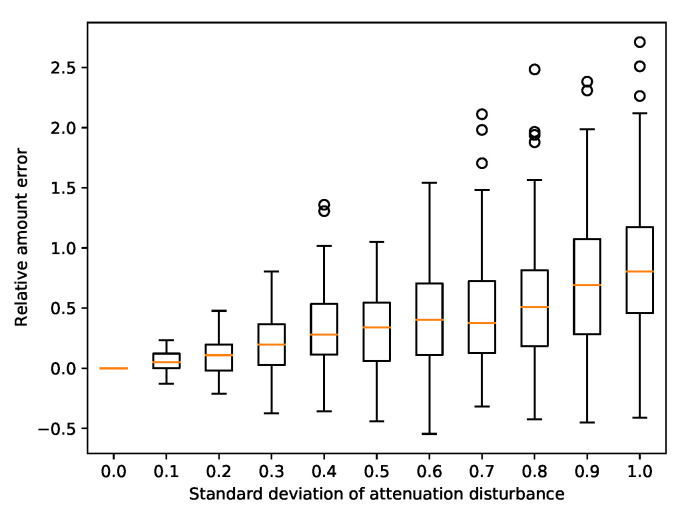
Amount error versus attenuation disturbance.

**Figure 12 sensors-22-00387-f012:**
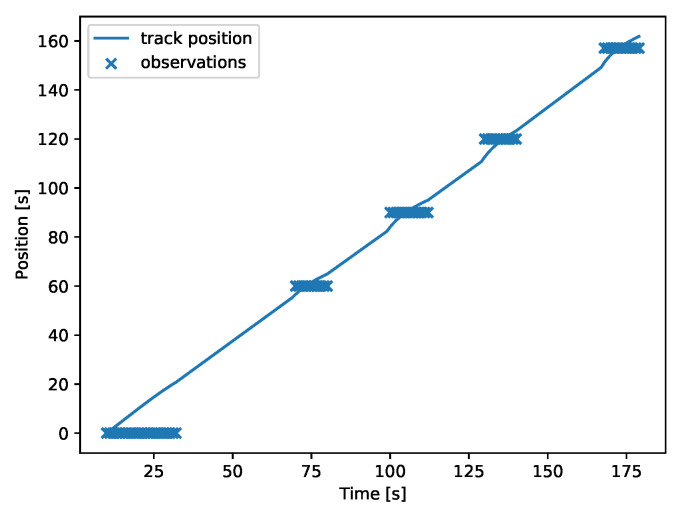
Doubling of signals. The target flows through two different paths of different lengths, thus producing five peaks in a 4-node network.

**Figure 13 sensors-22-00387-f013:**
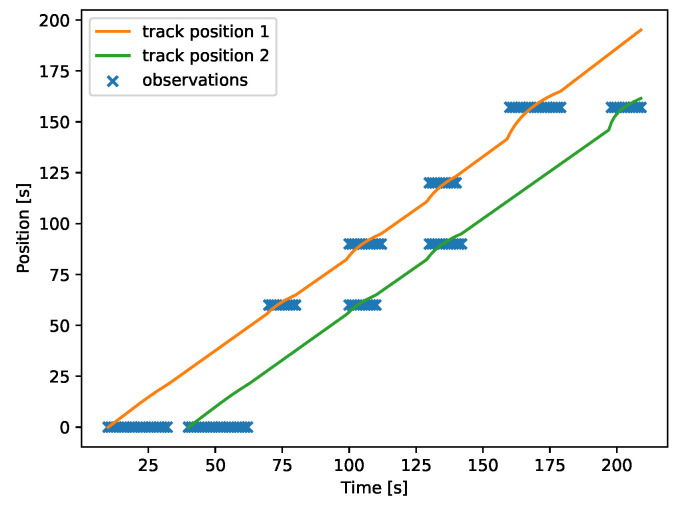
Two discharge events are correctly separated by the data fusion algorithm.

**Figure 14 sensors-22-00387-f014:**
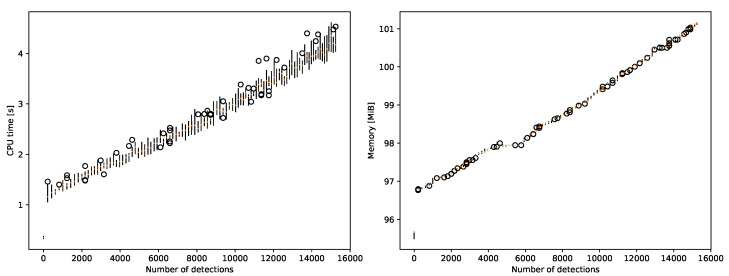
CPU time and memory usage depend on detection count.

**Figure 15 sensors-22-00387-f015:**
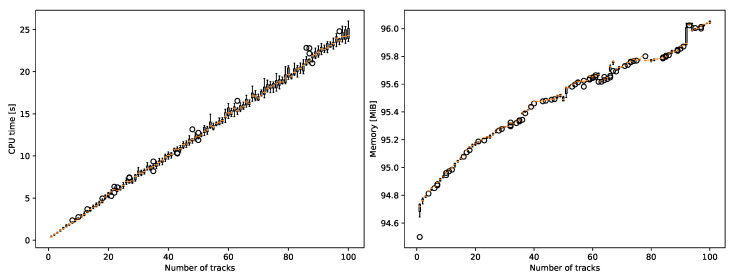
CPU time and memory usage depend on track count.

## Data Availability

Restrictions apply to the availability of these data. Data were obtained from Steffen Krause and Christoph Wöllgens of the Universität der Bundeswehr München and are available from the authors with the permission of Universität der Bundeswehr München.

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
