# Peer review of "Multisensor Data Fusion for Localization of Pollution Sources in Wastewater Networks"

_sensors, 2022, doi:10.3390/s22010387_

Round 1

Reviewer 1 Report

This manuscript provides an enhanced algorithm for multi-sensor data fusion for the detection, localization, and quantification of pollutants in wastewater networks. The problem is interesting. However, there are some problems with the manuscript.

  1. The necessity and evidence of proposing the enhanced data fusion framework, which consists of resampling, peak detection, pollution quantification, downstream propagation, tracking, and event generation are not provided.

The introduction should be explained in more depth.

The hierarcical framework to apply in the pollutants estimation is popular. The authors should discuss state-of-the-art with more detail.

  1. In line 188, is the object’s output, or in this case, the observable state in time. What exactly does stand for?

The process and observation noise are not considered in Equation (9) and Equation (10). Please clarify the noise. The authors may enhance the description of the Kalman filter process for tracking in DAGs in more depth.

  1. The results validation section is weak. To justify its advance, the authors should compare the proposed algorithm with other methods, such as the authors' previous work (reference [1]).

Reviewer 2 Report

The addressed topic is very interesting, the authors aim to propose an algorithm based on multisensor data fusion for the detection, localization, and quantification of pollutants in wastewater networks. Although the results are promising there are several issues that must be addressed.

  1. The Introduction lacks of relevant information, that is, in the Introduction section is commonly presented the addressed problem, previous advances and discussions about possible solutions.
  2. A detailed description of the proposed method must be given. Thus, the current version of Section 2 "Methods" must be include more details, also, Figure 1 can be used to provide a detailed and visual description of the proposed method.
  3. It is not clear how the datafusion is performed, in Line 81 is mentioned that "The data fusion algorithm is a loop of six steps:", but the fusion process is not defined.
  4. The result have to be compared with results achieved by applying other similar related methods, through this comparison the authors have to highlight the advantages of this proposal.
  5. The Conclusions are not convincing, please re-write the Conclusion section

Reviewer 3 Report

The manuscript by Chachula et al. titled, “Multisensor Data Fusion for Localization of Pollution Sources in Wastewater Networks” proposes improvements over a previously published approach on multisensory data fusion for wastewater network pollutant detection, tracking, and notification. The study incorporates an algorithm to do this and provides metrics of success and accuracy of this algorithm. I appreciate the authors’ effort in addressing this problem across different synthetic and real networks and analyzing the performance dependencies on the state of the system. Overall, this is a good study but requires additional information and analysis before publication, especially quantifying the improvement when compared to the previously published research, Chachula et al. 2021 (https://doi.org/10.3390/s21030826).

The introduction section does not do a good job in identifying the problem at hand and establishing the need for a solution which reduces the impact of the work. It is important to establish novelty of the work in the introduction section.

Abstract Line 1: Please refer to where this is a growing problem.

Please include the statistics for the measure of quality of event detections and performance and resource usage of the system in the abstract itself.

  1. Introduction: Please provide reference for the first and second sentences of the introduction. Also include discussion on why localization and quantification are immensely challenging. The introduction does not provide enough information as to why the problem the authors are solving is important and directly goes into relevant work.
    • Related Work

This section just lists the previous studies without establishing why these are included in the text. What were the advantages and disadvantages of these studies that will be addressed from the work of the authors? This again goes back to my point about the description on the need for this study. What equations are used to estimate the gradual dilution of the pollutant in flowing water? Please comment on how the choice of equal weights to the nodes might cause inaccurate simulations. What is the uncertainty of sensor measurements? How would this uncertainty be incorporated into the sensor monitoring accuracy?

  1. Methods

The dispersion factor: Please provide reference of why this approach is used to calculate dispersion of the pollutant.

How much error is introduced into the flows in the system due to resampling?

Figure 1. A better figure with image insets is needed to better understand this process. The description in the methodology requires more details as to why these steps are followed.

  • 2.1 and 2.2 The previous study by the authors Chachula et al. 2021 (https://doi.org/10.3390/s21030826) addresses a very similar problem but this study provides better solutions to some of the problems from the first study. A discussion of the previous study should be included in the introduction itself. Also, quantification of the overall improvement in accuracy between the proposed approach and the previously published approach needs to be provided to statistically conclude that the new approach is indeed better. Why is the velocity in a track is constant? Why was a better approach not used for this assumption?
  1. Results

It is important to compare performance accuracy with the previously proposed approach. Using the same EPANET network in addition to the other three proposed would enable that comparison.

3.5 Sensor distance error: Maybe this analysis can be used to quantify the uncertainty induced by sensor placement /amount in the system?

3.6 Figures 7, 8, 9 and 10 can be presented much more clearly. It is difficult to understand with the current format.

3.8 Why do the authors choose to not model the distribution of flow in the system using a Poisson distribution as it is the standard for such networks? Please elaborate and list in the limitations of the study.

4. The conclusions section should list all the limitations of the study and discuss possible implications.

Round 2

Reviewer 2 Report

The manuscript has been significantly improved, all comments were attended by the authors.

However, the sensor data fusion is one most relevant contributions that is intended to be reported; in this sense, the importance of performing sensor data fusion must be highlighted. Also, it could be mentioned that sensor data fusion and/or multiple signal data fusion may lead to achieve high-performance results in a wide-range of application, i.e., condition monitoring.

https://doi.org/10.3390/s21175832

https://doi.org/10.1109/TIA.2016.2637307

Finally, please don't write the manuscript in a personal way, all phrases like "we proposed" must to be changed by "In this work is proposed".

Author Response

> The manuscript has been significantly improved, all comments were attended by
> the authors.

Thank you for all the comments and for your recognition of the improvements.

> However, the sensor data fusion is one most relevant contributions that is
> intended to be reported; in this sense, the importance of performing sensor
> data fusion must be highlighted.
> Also, it could be mentioned that sensor data fusion and/or multiple signal
> data fusion may lead to achieve high-performance results in a wide-range of
> application, i.e., condition monitoring.
> https://doi.org/10.3390/s21175832
> https://doi.org/10.1109/TIA.2016.2637307

Thank you for pointing this out and for providing additional references.
We emphasized the importance of performing data fusion by adding the following
paragraph to the introduction.

The need for data fusion in this field is supported by the fact that a simple
monitoring system where anomalies are detected independently for each sensor
would be lacking at least four desirable features.
The first one is marking discharges observed by multiple sensors as more
significant than those detected by only a single sensor.
The second is the quantification of pollutants by considering the sum of
deviations in measurement series connected to a single discharge event.
Another one would be the deduplication of alarms by clustering similar
observations.
The last one is discovering the flow path of the pollutant that leads to easier
source localization.
Data fusion has also proved to be a viable technique for knowledge discovery in
other fields such as bearing fault identification [23].

> Finally, please don't write the manuscript in a personal way, all phrases like
> "we proposed" must to be changed by "In this work is proposed".

Please note that many scientific papers that we cite use active voice.
It makes the text more clear, as perceived by some reviewers of our papers.
Forms "we propose", "we developed", "we applied", "we report", "we combine",
"we achieved" etc. are common.
Please see the following references: [3,5,8-11,15,17,18,27-32,34,36,38-40]

Reviewer 3 Report

I have thoroughly reviewed the revised version of the manuscript submitted by the authors and I am satisfied with their responses to the questions raised in the first review. There are a few grammatical errors that would require thorough proof reading. Therefore, I recommend accepting this manuscript after a minor revision.